# Point-of-Need DNA Testing for Detection of Foodborne Pathogenic Bacteria

**DOI:** 10.3390/s19051100

**Published:** 2019-03-04

**Authors:** Jasmina Vidic, Priya Vizzini, Marisa Manzano, Devon Kavanaugh, Nalini Ramarao, Milica Zivkovic, Vasa Radonic, Nikola Knezevic, Ioanna Giouroudi, Ivana Gadjanski

**Affiliations:** 1Micalis Institute, INRA, AgroParisTech, Université Paris-Saclay, 78350 Jouy-en-Josas, France; furezp@hotmail.it (P.V.); devon.kavanaugh@inra.fr (D.K.); nalini.ramarao@inra.fr (N.R.); 2Dipartimento di Scienze AgroAlimentari, Ambientali e Animali, Università di Udine, 33100 Udine, Italy; marisa.manzano@uniud.it; 3Institute of Molecular Genetics and Genetic Engineering (IMGGE), University of Belgrade, 11000 Belgrade, Serbia; milicanikolic@imgge.bg.ac.rs; 4BioSense—Research and Development Institute for Information Technologies in Biosystems, University of Novi Sad, 21000 Novi Sad, Serbia; vasarad@biosense.rs (V.R.); nknezevic@biosense.rs (N.K.); ioanna.giouroudi@biosense.rs (I.G.)

**Keywords:** biosensor, point-of-need DNA detection, DNA/RNA extraction, DNA amplification, microfluidic, food security

## Abstract

Foodborne pathogenic bacteria present a crucial food safety issue. Conventional diagnostic methods are time-consuming and can be only performed on previously produced food. The advancing field of point-of-need diagnostic devices integrating molecular methods, biosensors, microfluidics, and nanomaterials offers new avenues for swift, low-cost detection of pathogens with high sensitivity and specificity. These analyses and screening of food items can be performed during all phases of production. This review presents major developments achieved in recent years in point-of-need diagnostics in land-based sector and sheds light on current challenges in achieving wider acceptance of portable devices in the food industry. Particular emphasis is placed on methods for testing nucleic acids, protocols for portable nucleic acid extraction and amplification, as well as on the means for low-cost detection and read-out signal amplification.

## 1. Introduction

Bacterial contamination of food and water is a crucial food safety issue as it is linked to increased mortality, human and animal suffering and economic burden. Outbreaks caused by bacteria share a common symptomatology (diarrhea, fever, vomiting), leading to difficulties in identification of the responsible causative agent. The significant public concern about the safety of food and water highlights the need for tighter monitoring of pathogens. The economic importance of these practices is illustrated by the high value of the European food safety testing market, dominated by pathogen testing, which reached $4 billion in 2018 and is expected to reach $6.5 billion by 2025 [1].

The global incidence of foodborne disease is difficult to estimate. The European Food Safety Authority and the European Centre for Disease Prevention and Control have reported approximately 359,700 hospitalizations due to confirmed zoonoses and nearly 500 fatal cases in EU in 2016 [2]. The US Center for Disease Control and Prevention (CDC) 2011 Estimates attributed as many as 128,000 hospitalizations and 3000 deaths in US annually to the contamination of food and drinking water. It is worth noting that the true incidence of foodborne outbreaks is highly underestimated for many reasons, among which, misdiagnosis, under-reporting (particularly of minor outbreaks), and improper sample collection and testing. 

Among bacterial infections salmonellosis, Campylobacteriosis and STEC infections are responsible for the vast majority of illnesses, hospitalizations, and deaths. Besides, although Listeriosis is not frequent, because *Listeria monocytogenes* has a lower prevalence in foods compared to *Campylobacter* and *Salmonella*, it is one of the most potent foodborne diseases, with an associated mortality rate reaching as high as 37% [2]. Traditional methods for bacterial detection and enumeration from food matrices are based on bacterial culture on agar plates. The initial results take 2–3 days, while confirming the specific pathogen may take more than one week. Conventional microbiological methods based on culture and colony counting, thus, do not meet the demand of rapid food testing. Time requirements needed to provide final results is a crucial parameter in detecting certain foodborne pathogens as *Escherichia coli*, a leading cause of death in young children [2]. 

Some foodborne diseases are caused by bacteria that produce toxins. As a whole, EU-wide, foodborne toxigenic bacteria are on the rise and are considered as emerging threats. Of particular importance is the fact that genes coding for the toxins can be transferred among bacteria. More than 5000 food-borne outbreaks are reported annually in the EU (EFSA, 2017) [2], with bacterial toxins produced by *Staphylococcus* spp., *Bacillus* spp. and *Clostridium* spp. as causative agent, accounting for nearly 10,000 cases each year. Food-borne outbreaks caused by toxigenic bacteria often share a common symptomatology, thereby hindering outbreak investigations. In addition, adequate methods for bacterial toxin detection are lacking. Consequently, the proportion of “weak evidence” foodborne outbreaks is particularly high in case of bacterial toxins acting as the causative agent. Furthermore, detection of bacterial toxins is exceptionally important because toxins may remain in or on the food and be ingested while the bacteria is eliminated or no long present. The methods of toxin detection primarily rely on immunological assays such as ELISA, lateral flow immunoassays, and agglutination tests. In some cases, toxins are evidenced by bioassays in tissue culture, or mouse neutralization testing, and other live animal tests, all of them time consuming with some being ethically challenging. 

An emerging branch of analytical methods for pathogen detection with the potential to address weaknesses of classical methods combines biosensors, microfluidics, and nanotechnologies (Figure 1). Over the last decade, the field of portable sensors for food and water quality control has grown exponentially [3]. This review aims to provide a focused overview of point-of-need biosensors for DNA detection, paying particular attention to recent trends in nucleic acid extraction, amplification, and amplicon detection principles.

## 2. Classical Methods for Foodborne Pathogen Detection

The conventional methods implemented in food analysis consist of sample homogenization and subsequent culturing of the microorganisms on agar plates followed by biochemical identification [4,5]. The plate count method, developed decades ago, is still used in various official protocols for microbial enumeration (Figure 2). The method requires specific media for enrichment, isolation and identification of the microorganism to provide the number of viable bacterial cells in a sample. In addition, selective culture media of a defined composition, specific atmosphere (oxygen content), and optimized incubation temperatures are required for different bacterial strains to grow. While the method is specific, it is also time consuming and costly since a high number of Petri dishes, media, plastics for the analyses as well as trained personnel are needed. Moreover, it may take up to one week to successfully determine contamination by pathogens present in low numbers in a food sample, such as *L. monocytogenes, Salmonella* spp., or *Campylobacter* spp. For instance, the official ISO 11290 method for detection of *L. monocytogenes* is based on two enrichment steps in broth before bacterial plating onto selective agar medium, followed by the confirmation test. Other cell culture methods have been proposed for microbial enumeration in recent years. The most common means of enumeration of live bacteria, is the use of serial dilution of growth culture. This is often used for qualitative or semi-quantitative analysis, with samples that are belived to contain compounds that can interfere with plate count methods. This method is laborious as it requires 3–5 replicates for each dilution to obtain reliable results. 

The emergence of polymerase chain reaction (PCR) has changed the way of performing microbiological analyses towards the detection of specific microbial DNA as a target (Figure 3). Some pathogens, like many *Salmonella* and *Campylobacter* strains, may be viable but non-culturable (VBNC). Using culturing methods for their detection leads to a false negative result and a failure in pathogen detection. Molecular PCR-based methods that detect pathogen-derived nucleic acid (DNA or RNA) prevent this risk. A plethora of PCR-based methods has been developed for various purposes including: nested PCR, multiplex PCR (mPCR) and real time PCR (RT-PCR). PCR typically relies on the annealing of a pair of primers specific for DNA template in question. PCR technique is widely employed in food safety analysis for both speed and simplicity of utilization. Moreover, many kits have been developed to facilitate food testing. Classical end-point PCR gives only a qualitative result, while real-time PCR can provide the number of pathogen cells present in a given food sample. Unfortunately, PCR-based detection of pathogens requires a pre-enrichment step to increase the number of cells for detection and to eliminate the risks of detecting DNA from dead bacteria. New protocols tackle the problem by adding cell membrane-impermeable dye to PCR reagents that can penetrate only into dead cells (such as propidium monoazide). The dye can bind only to the extracellular DNA (e.g., passively released from dying cells), hence selectively preventing PCR amplification of DNA from the dead cells [6]. This ensures that only foods contaminated with living bacterial cells produce an amplicon. 

Real-time PCR or quantitative PCR (qPCR), does not require agarose gel electrophoresis to provide a result as the formation of PCR products is continuously monitored by measuring a fluorescent signal produced by the continuous production of the amplicons [7]. Commonly, SYBR green, a dsDNA-binding fluorescent dye is used, but also hydrolysis probes, as TagMan, and molecular beacons can be employed for this purpose. TaqMan probes, which are complementary to a specific nucleotide sequence in one of the amplicon strands, contain a fluorophore as the reporter at the 5′-end and the quenching molecule at the 3′-end [8]. The molecular beacons are probes containing a hairpin/stem-and-loop configuration beginning with a quenched fluorophore which becomes ‘unquenched’ producing fluorescence after annealing of the probe to the complementary nucleotide sequence of the amplicon [9]. 

A multiplex qPCR assay has been developed for the detection and quantification of multiple foodborne pathogens by Fratamico et al. [10] and Hu et al. [11]. The pathogens targeted by the assay include *Salmonella enterica subsp. enterica, Listeria monocytogenes, Escherichia coli O157, Vibrio parahaemolyticus, Vibrio vulnificus, Campylobacter jejuni, Enterobacter sakazakii* and *Shigella* spp. For seafood, meat, and ready-to-eat product analysis, TaqMan multiplex real-time PCR method for the detection of *Salmonella*, *Shigella* and *L. monocytogenes* was developed, although enrichment medium is also required for the simultaneous growth of the bacteria of interest. The limit of detection (LoD) for the three bacteria analyzed was established between 3 and 22 colony forming units (cfu)/25 g of food sample [12]. Notably, multiplex qPCR can be used for multiple pathogen detection in the presence of high numbers of contaminating microorganisms [13,14]. Also, Zhou et al. [7] developed a mPCR for the simultaneous detection of six pathogens: *Salmonella enterica*, *Listeria monocytogenes*, *Staphylococcus aureus, Escherichia coli O157:H7, Shigella* spp. and *Campylobacter jejuni.* The LoD of these methods, which also depends on the extraction methods used, was similar to those obtained by traditional culturing methods or slightly lower (approximately 10 cfu/25 g). Nevertheless, in the case of samples analyzed with qPCR prior to the enrichment step, the detection limit was in the range of 10^2^–10^3^ cfu/g (or mL). The main advantages of molecular over culture-based methods are the shorter time required to obtain reliable results, and the possibility of simultaneous detection of several targets.

## 3. Detection of Bacterial Toxins

As mentioned above some bacteria, such as *Clostridium botulinum*, *C. perfringens*, *Staphylococcus aureus*, *Bacillus cereus*, Shiga-toxin-Producing *Escherichia coli* (STEC), and *Vibrio parahemolyticus*, may produce toxins which cause foodborne illnesses. For example, *Staphylococcus aureus* produces Staphylococcal enterotoxin which causes a form of food poisoning. Some toxins, including Staphylococal SEB or *B. cereus* cereulide, are heat resistant and persist on surfaces and food matrices for significant periods of time [15,16]. Staphylococcal food poisoning is characterized by nausea followed by incoercible vomiting caused by the ingestion of food contaminated with heat-stable staphylococcal enterotoxins. Despite the fact that there has been a food safety criterion for staphylococcal enterotoxins put into force according to EU legislation No. 2073/2005, detection and quantification of staphylococcal enterotoxins in food remnants remains a real challenge, with no “gold standard” currently available to perform unambiguous quantification of enterotoxins in foods. The available immunoenzymatic methods allows the detection of only five out of 24 staphylococcal enterotoxin serotypes (namely SEA, SEB, SEC, SED and SEE). In particular, epidemiological data indicate that there is an increase of cases caused by emerging *S. aureus* strains producing other types of toxins such as SEG, SHE and SEI.

*E. coli* O157: H7 produces Shiga-like toxins 1 and 2 that causes dysentery, haemorrhagic colitis, and haemolytic uremic syndrome. The recent European O104:H4 outbreak was caused by a typical enteroaggregative *E. coli* strain that had acquired the bacteriophage encoding *Stx* (Shiga toxin). The high persistence of *Stx* phage up to 1 month at pH 7–9 indicates that it is not enough to detect only bacterial strains, but also their toxins [17]. As such, STEC virulence factors encoded on mobile DNA elements could spread among other pathotypes of diarrheagenic *E. coli* and thereafter represent a public health threat [18]. For *Stx* in particular, the CDC (U.S.) published guidelines in 2009 stipulating that clinical laboratories should perform simultaneously an assay that will detect either the Shiga toxins directly or the genes encoding them and a selective culture for O157 STEC [18].

*Clostridium perfringens* gastroenteritis is caused by the production of *C. perfringens* enterotoxin encoded by the *cpe* gene. Recently, additional genes with a toxigenic potential have been identified in *C. perfringens* strains, however, functional data about their role in development of gastroenteritis are lacking. The detection of the *C. perfringens* toxins is currently based on the detection of their genes by PCR.

*Bacillus cereus* may cause an emetic or a diarrheal-type of food-borne illness [19]. Diarrheal strains produce toxins such as Hbl and Nhe that will induce the symptoms following their production in the gut [20,21]. Emetic strains produce the emetic toxin, encoded by the *ces* gene, a heat stable toxin that is preformed in the food. The ingestion of this toxin results in the ensuing intoxination. Proteomics approaches based on LC-MS/MS experiments (either on intact or trypsin-digested proteins) and mass spectrometry methods have been developed to detect *B. cereus* toxin in various food matrices. As these toxins may be present in food in the absence of bacteria, their detection is a prerequisite for food safety.

The most widely used means of direct detection of bacterial toxins are ELISA or RPLA (reversed passive latex agglutination) tests, for which commercial kits have been developed. To bypass issues of low toxin concentration, kits may include immunoaffinity columns for the concentration of toxins. These tests generally require 4–24 h to provide a result. Further refinements have yielded lateral flow test systems which benefit from the use of monoclonal, rather than polyclonal, antibodies for improved specificity and are promoted as suitable for the rapid (time to result 4–6 h) screening of food samples as well as for confirmation of bacteria in culture. Recently, detection has begun to include the use of mass spectrometry. Mass spectrometry has been successfully used to identify the seven serotypes of botulinum toxin and the lethal toxin of *Bacillus anthracis* [22] meriting the inclusion of the technique into previous European frameworks aiming to identify and detect pathogen-associated toxins [23]. Another technology successfully performed to detect *Clostridium difficile* toxins is single-molecule array technology utilizing paramagnetic beads coated with antibody [24,25]. Detection of bacterial toxins generally focuses on direct recognition of the toxin itself, either by means of antibody recognition or interaction with a substrate. In addition, toxins can be also detected by the presence of the toxin-encoding gene [26,27]. Recent advances have been made regarding whole genome sequencing. These technologies are now rapid, efficient and cheap, enabling the acquisition of information on the toxigenic potential of a given strain.

Each of the aforementioned methods for bacterial toxin detection possesses advantages and disadvantages, often trading practicality and ease of use in exchange for specificity, limit of detection, and/or cost. Given the variability among bacterial strains and their toxins, as well as their adaptability, the field of toxin detection is required to continually evolve. For example, incorporation of nanomaterials into detection methods shows significant promise. Another example for advancement of detection methods is the use of DNA probes instead of antibodies. Antibodies are frequently used in the design of ELISA assays for the detection of whole bacteria or their toxins, but they also bear an inherent risk of cross-reactivity, thereby potentially reducing their specificity. In contrast, DNA probes, when designed carefully, enable the highly specific detection of bacteria or the presence of their toxin-producing genes. 

## 4. Point-of-Need Detection of Specific Nucleic Acid Sequences

There is an acute need in the land-based sectors (agriculture, environmental, plant and animal care sector) for novel rapid on-site i.e., point-of-need biosensing systems for early detection of zoonotic diseases and for freshwater monitoring with high sensitivity and specificity. Biosensors harness the specificity and sensitivity of biological systems in small, low cost devices providing a powerful alternative to conventional methods [5,28]. Moreover, conventional food analysis for the detection of harmful bacteria and their toxins can be only performed on the previously produced food [29,30]. Biosensors, however, enable analysis and screening of food items during all phases of production providing more adapted and more efficient tools to ensure of food safety.

Current biosensor research is directed towards integration of Nucleic Acid Testing (NAT) into microfluidic devices to further increase the biosensing capacity and develop diagnostic tools that can meet Affordable, Sensitive, Specific, User-friendly, Rapid and robust, Equipment-free and Deliverable (ASSURED) criteria recommended by the World Health Organization (WHO) [31,32,33]. Use of NATs enables swift detection of pathogens as the complete test is minimized to several hours. The first step in a nucleic acid detection is the extraction of pathogenic DNA/RNA from the food matrix. The second step is detection of the specific nucleic acid sequence, which usually demands a pre-amplification step. The most popular laboratory methods for nucleic acid amplification are still PCR, RT-PCR and qPCR. However, PCR requires thermal cycling to obtain the target sequence amplification. The need for a thermal cycler instrument significantly limits the potential for PCR integration into microfluidic devices for point-of-need testing. On-going research studies strive to overcome these technical challenges by achieving nucleic acid sequence amplification through the use of sensitive isothermal methods which operate at constant temperature and greatly simplify the entire process of amplification, since they make use of enzymes to perform strand separation that would otherwise require repeated heating (thermal cycling) [34]. Isothermal NATs are also better suited for integration into microfluidic devices. 

In addition, the hardware component of signal detection can be incorporated into the microfluidic chip, providing a full ‘lab-on-a-chip’ on-site detection device. The general trend for point-of-need applications is to utilize simple and cheap hardware components, such as LED diodes and battery-free RFID (radio-frequency identification) antennas [35] and make use of smartphones for signal quantification and visualization [36] (see the section Detection methods). Confinement of the sample into microfluidic environment of the device reduces the risk of sample contamination and minimize the sample volume and reagents required for the assay, thereby further decreasing the overall cost of screening and detection.

The continuing development of both conventional and isothermal nucleic acid testing methods is primarily oriented to healthcare applications e.g., for infectious diseases, diagnostics, early cancer diagnostics, detection of genetic biomarkers of tumors and drug metabolism, and analysis of extracellular vesicle content. NATs have been studied particularly extensively in paper-based diagnostics, presenting substantially higher sensitivity and specificity than immunoassays [31,37]. Additionally, NAT-based analytical methods find their place in agriculture and other land-based sectors as well as by enabling fast and accurate detection of contamination of food and the environment [38,39].

In general, the procedure for rapid detection of DNA/RNA from pathogenic bacterial comprises five steps: pre-concentration, extraction, detection, signal transduction into a measurable signal and data analysis. In addition, the procedure can be complemented with on-site DNA sequencing using new technology for fast sequencing developed in 2014 by Oxford Nanopore Technology (Oxford, UK). This portable method is based on the utilization of a USB-powered sequencer (MinION) that comprises thousands of wells, each containing nanopores. When DNA enters the pore, each base produces a unique electronic signature that can be detected by the system, providing a readout of the DNA sequence. The combination of isoNATs with portable DNA sequencing has been heavily investigated for human infectious diseases, such as malaria [40] or dengue virus [41] and is rapidly gaining attention for point-of-need detection of foodborne pathogens as well [42].

## 5. Extraction Methods

Extraction of DNA is a crucial step for successful pathogen detection by a molecular method, since the reproducibility and sensitivity of detection directly depends on the purity and integrity of the DNA. A general guidance for extraction and quantification of genomic DNA is provided by the ISO standard 21571 annexes (ISO 21571:2005). Various optimizations of DNA extraction for specific applications have resulted in a plethora of methods and protocols. They all contain, as a first step, liberation of the bacteria from the starting material, followed by the lysis of bacterial cells by disruption of their membranes and cell walls. The next steps comprise separation of DNA from proteins and cell debris and purification of the extracted DNA. There is no universal rule and, typically, the extraction method has to be adapted and optimized to a given food matrix and a given bacterial agent. Classical extraction procedures are time consuming and often cannot be miniaturized because of the successive filtration and centrifugation steps. When applied to food sample analysis, classical extraction methods may fail to yield good quality DNA due to the complexity of the food matrix that can contain ions and molecules able to inhibit enzymes used for amplification. Use of an adequate lysis buffer helps to optimize an efficient DNA extraction procedure. Tagliavia et al. [43] used a lysis solution containing KOH, Na_2_EDTA and Triton, to perform an efficient and rapid extraction of DNA from fresh, stored and processed seafood. Similarly, a comparison of several commercial kits as well as extraction methods based on urea/SDS/proteinase K, phenol/chloroform and salt demonstrated that all tested procedures were suitable to successfully extract DNA from fish muscle tissues [44]. Recently, Torelli et al. [45] reported a simple and fast DNA extraction protocol based on proteinase K and lysis buffer containing Guanidine-HCl, EDTA, NaCl, Triton and SDS. This lysis buffer allowed the extraction of DNA from meat samples in only 30 min compared to 5 h needed with a commercial kit [45]. The lysis buffer improved the disruption of the cell of the meat sample enabling the successful DNA hybridization to a specific probe during the subsequent detection step.

The phenol-chloroform method is an efficient method for DNA extraction from many microorganisms [46]. A comparative study of DNA extraction methods of pathogenic bacteria concluded that several simple, rapid, and affordable physical methods can yield DNA of good quality when applied on pure bacterial culture, but that the phenol/chloroform method is superior to physical methods when bacterial DNA is extracted from beef [47]. 

The quality of DNA extracted from processed foods represents a challenge, as since obtained DNA may be of poor quality and in low quantity. Chemical and thermal treatments of food result in fragmentation and random breaks of long DNA strands, consequently making their use in PCR very difficult. However, DNA fragmentation is more of an obstacle concerning food origin characterization that requires whole stretch of DNA, as opposed to pathogen detection, which can be performed with short DNA segments. To improve extraction of DNA from processed foods, the cethyltrimethylamonium bromide-mediated (CTAB) method is frequently used. The basic protocol using CTAB extraction buffer and chloroform (or isoamyl alcohol) to remove proteins and polysaccharides from DNA was modified by the addition of various salts, SDS, proteinase K, or mercapto-ethanol to obtain DNA free of PCR-inhibitory compounds [48]. The extraction procedure has a strong impact on the limit of detection of qPCR. Various food matrix compositions require specific treatments/reagents to exclude inhibitors of amplification enzymes. However, when applied to milk and dairy products the LoD was found to deteriorate with increasing of fat content. Using Lambda DNA as an internal control, the authors demonstrated the presence of a qPCR-inhibiting molecule in the milk fat [49]. Pirondini et al. [50] confirmed the suitability of the method using CTAB for genomic DNA extraction from dairy foods when compared with a number of commercial kits or protocols using Tween or SDS. CTAB-extracted DNA from seeds and grains requires additional purification step to remove polysaccharides which may inhibit q-PCR [51]. Yalcinkaya et al. found the salt method which uses a high concentration of NaCl (6 M) easy to perform, inexpensive, and environmental-friendly when applied to DNA extraction from beef samples [52]. Furthermore, compared to CTAB, alkaline, Tris-EDTA, urea or guanidium isothiocyanate methods, the salt method provided the highest yield of DNA when applied to meat and meat products. 

## 6. Pre-Concentration Methods

The efficiency of an extraction method possessing pathogen as a target can be improved with a pre-concentration step that precedes extraction. Adding a pre-concentration step to the extraction procedure may be laborious and expensive, but the pre-concentration step may prove to be necessary to provide an optimal DNA quantity for pathogens present at low numbers in food samples. In addition, multiple pathogens can be present in the same food sample. For instance, chicken contaminated with *Campylobacter* is usually co-contaminated with *E. coli*. A pre-concentration step provides enrichment and isolation of the given pathogen prior to analysis. The possibility to miniaturize and automate pre-concentration protocols based on functionalized magnetic beads with antibodies/aptamers/bacteriophage-derived proteins will enable the development of point-of-need kits. 

Amoako et al. [53] showed efficient immunomagnetic separation in combination with pyrosequencing detection of *B. anthracis* spores in liquid foods (bottled water, milk, juice), and processed meat experimentally inoculated with anthrax spores, without an enrichment step. Magnetic beads decorated with specific anti-*B. anthracis* antibody were used to separate spores of *B. anthracis* from the food samples. Although some kits for detection of anthrax spores were found to be quite efficient on their own, the authors obtained the best result with the selective pre-concentration of spores owing to the fact that immunomagentic separation provided DNA free of qPCR-inhibitors [53]. Fischer et al. demonstrated that pre-concentration of *B. cereus* spores from milk with specific aptamers attached to magnetic beads provided a cost-efficient trapping method for routine analysis in the food industry without the need to perform a time-consuming enrichment step [54]. The aptamer-based trapping of *B. cereus* enabled subsequent bacterial identification by real-time PCR in both milk-simulating buffers and milk of varying fat content. Vinayaka et al. demonstrated effective real-time PCR detection of low levels of *Salmonella enterica* without culture enrichment, using antibodies immobilized on magnetic beads. The capture efficiency was 95%. In direct PCR, a strong linear relationship between bacteria concentration and the number of cycles was observed with a relative PCR efficiency of ~92% resulting in a limit of detection of ~2 cfu/mL [55]. 

Bacteriophage or bacteriophage receptor binding protein can be used for highly-specific bacterial separation from food matrices, as they have excellent selectivity for a specific bacterium [56]. Their ability to infect bacteria in a strain-specific manner enables development of specific analytical platforms to separate bacteria from a food sample. Bennett et al. [57] reported for the first time the use of bacteriophage-based systems for separation and concentration of *Salmonella* and *E. coli* in a mixed-broth culture. Walcher et al. [58] coupled phage cell-wall domains to paramagnetic beads and used them to capture *L. monocytogenes* from milk. Poshtiban et al. [59] demonstrated the use of immobilized recombinant phage receptor binding proteins, responsible for the phage-host specificity, onto magnetic particles for *C. jejuni* cell isolation from food samples in less than 3 h. With an estimated pool of 10^31^ phages existing in the environment, bacteriophages provide a unique class of separation elements.

Highly performing automated pre-concentration step is particularly important for bacterial detection by portable biosensors as it increases bacterial concentration and reduces the volume to analyze. Typically, less than 1 cfu/mL of bacterial pathogen is present in food and contaminated water. Despite the seemingly low numbers, these quantities of bacteria with low infectious may cause a serious threat to human and animal health. At the same time, these low concentrations are often below the LoD of most devices, leading to false-negatives during identification. Consequently, many portable devices cannot be approved to replace conventional methods of analysis. For example, the Food and Drug Administration (FDA) acceptable limit for *L. monocytogenes* in ready-to-eat food products is <1 cfu/25g of foods that support the growth of *L. monocytogenes*. Therefore, automated pre-concentration steps appears to be a suitable solution for in field applications of point-of-need devices in the food industry. 

## 7. Portable Extraction Methods

Combining magnetic micro- and nanoparticles with microfluidic systems enables on-chip DNA extraction that can be integrated into a portable lab-on-a-chip device for food molecular analysis. Silica-coated microstructures were integrated into a microchip containing both, a DNA extraction and purification componen, resulting in PCR-based rapid detection of *L. monocytogenes* in a single device [60]. This enabled detection of *L. monocytogenes* with an average turnaround time of 45 min. Govindarajan et al. [61] developed a low-cost paper microfluidic device for point-of-need extraction of bacterial DNA from raw viscous samples. The system contained a storage pad carrying a dry lysis buffer activated by addition of the sample utilizing a “microfluidic origami”. This on-chip platform provided cell lysis and *E. coli* DNA extraction from spiked pig mucin without the use of external power at room temperature, in only 1.5 h. Recently, Hugle et al. [62] developed a microfluidic chip combining free-flow electrophoretic pre-concentration of *E. coli* cells, with thermoelectric lysis of bacteria and gel-electrophoresis for purification of nucleic acids. The integration of these three steps in a single chip enabled fast and easy extraction of bacterial DNA without the need for laboratory facilities.

To overcome limitations of conventional methods for the extraction of DNA, Tang et al. [63] have developed a paper-based DNA extraction device by incorporating a sponge-based buffer storage and paper-based valve and channels of different length to extract and capture DNA (Figure 4). The disposable device enabled automated DNA extraction within 2 min from only 30 μL of either contaminated biological samples or bacterial suspension. The performance was shown to be similar to that of the commercial DNA micro kits but simpler, inexpensive and as being portable and better adapted for point-of-need testing.

It is worth noting, that some amplification protocols can be performed without the need for nucleic acid extraction. For example, Williams and Hashsham suggested the use of direct, or DNA extraction-free amplification of pathogens from food matrices as an efficient way to reduce time to results in comparison to DNA extraction-based approaches. In their recent paper, they describe protocols for assay design of direct amplification using isothermal NAT and PCR for *E. coli* from milk samples and *Salmonella* from pork meat samples [64]. 

## 8. DNA Probe Design

DNA probes serve as sensing elements in biosensors as they hybridize with the extracted target nucleic acid sequences. The design of robust DNA probes requires certain criteria to be met, but when done properly, is considered the most specific means of bacterial detection. The basic objectives of probe design include (i) maximal specificity of probes; (ii) minimization of probe non-specific interaction; (iii) uniform probe melting temperatures, and (iv) reduction of secondary structure formation, which prevents probe access to targets.

Bacterial genomes require additional attention in the design of robust oligonucleotide probes. This is due to the low GC content and complexity in sequence composition, as well as common conserved repeats [65,66]. In general, the drawbacks associated with DNA or PCR probes include uncontrolled cross-hybridization on repeats, unpredicted secondary structures, or partial homology among regions of PCR probes. This can be further complicated by assay resolution variation among platforms, and the level of differentiation required, in relation to species within a limited number of families, or single strains in an even narrower range of species. These problems in oligonucleotide probe design have been addressed through the creation of several software packages to aid in their design and verification, including OligoWiz platform, ArrayDesigner 2.0, OliCheck, ORMA, ProbeMaker, ARB, and PathogenMIPer, to name a few. In the case of multiplex detection by long DNA probes (>40 base pairs) a uniform probe melting temperature is needed. The melting point essentially reflects the distribution of GC along the probe. Instead, with a shorter probe consisting of 20–40 base pairs, the uniform melting temperature is typically not necessary, as the range for the GC content is 40–60%.

DNA probe design begins with careful selection of starting sequences, relying on the analysis of similar, yet not-identical stretches of DNA, to enable a wider range of species to be differentiated by specific differences. As such, grouping of the sequences in clusters is recommended, thus maximizing the detection power and minimizing the crosstalk between the probes. There are several software platforms available to assist in the design of oligonucleotide probes. Once probes have been designed, BLAST is employed to determine similarity between probes and targeted and non-targeted subjects to further assist in identifying ideal candidates. Successfully designed DNA probes has led to many biosensor developments for applications in food safety, including different *Salmonella* serotypes [67], *Campylobacter* spp. [68], *L. monocytogenes* [69] or hepatitis A virus [70].

## 9. Portable Amplification Methods

Conventional NATs comprise polymerase chain reaction (PCR) and real-time PCR (RT-PCR) methods with high precision variants such as digital PCR (dPCR) [71] and droplet-digital PCR (ddPCR) [72]. However, due to the need for a duplex melting step during the PCR cycle, which requires heating, PCR is not appropriate for the detection of nucleic acid sequences for point-of-need testing applications. Isothermal NAT strategies, on the other hand, are performed at constant temperature, which simplifies the entire procedure, making it more applicable in the field. Some isoNAT methods can be performed at physiological temperatures (30–37 °C) or even room temperature [73]. Due to the simplified procedures for nucleic acid amplification, isothermal NATs also demonstrate a high potential for integration with microfluidic technology [28], and developing portable low-cost for point-of-need devices for molecular diagnostics with high sensitivity. Various isothermal amplification methods exist, such as loop-mediated isothermal amplification (LAMP), recombinase polymerase amplification (RPA), helicase-dependent amplification (HAD), rolling circle amplification (RCA), strand displacement amplification (SDA) and nucleic acid sequence-based amplification (NASBA) [74,75]. 

Among the currently available isothermal NATs, LAMP is the most widely researched method, offering significant support during the development process [76,77,78]. It is based on a set of four (or six) different primers that bind to six (or eight) different regions on the target gene making it highly specific [76]. LAMP has been employed for pathogen detection, as well as, in clinical diagnostics as it exhibited significant advantages, such as high sensitivity, specificity and rapidity [79]. The high sensitivity of LAMP enables the detection of pathogens in sample materials even in the absence of sample preparation [77]. The assimilating probes technique enables sequence-specific real-time monitoring of LAMP reactions directly in the reaction tube without subsequent molecular analysis [80]. This technique allows one-step application of LAMP with higher specificity, and significantly lowers the risk of contaminating subsequent reactions. The reaction can be monitored in real-time with an inexpensive handheld device, leveraging the simple LAMP process for mobile diagnostics and point-of-need testing [81]. Since LAMP can amplify a target DNA up to 10^9^ copies under isothermal conditions within tens of minutes, it has been successfully applied in detecting food borne pathogens such as *E. coli* in chicken meat [82], *V. parahaemolyticus* in seafood samples [83], or *C. botulinum BoNT/A* and *BoNT/B* genes in fish and honey [84]. 

Recombinase polymerase amplification (RPA) is recently becoming a molecular tool of choice for the rapid, specific, and cost-effective identification of pathogens [5,85,86,87]. The RPA process employs three enzymes: a recombinase, a single-stranded DNA-binding protein (SSB), and a strand-displacing polymerase. The recombinase is capable of pairing oligonucleotide primers with their homologous sequences in the target DNA. SSB then binds to the displaced strand of DNA and prevents the dissociation of primers. Finally, the strand displacing polymerase begins DNA synthesis where the primer has bound to the target DNA. With the use of two opposing primers, exponential amplification of the target sequence with RPA can be achieved at a constant temperature in 10–20 min. The RPA product can be measured in real-time using various probes [88]. Owing to minimal sample-preparation requirements, low operation temperature (25–42 °C), and commercial availability of freeze-dried reagents, this method has been applied outside laboratory settings, in remote areas, and onboard automated sample-to-answer microfluidic devices [89]. RPA also allows incorporation of fluorescent probes directly into the sample to be tested [90]. 

LAMP and RPA offer the option of multiplexing—parallel amplification and detection of multiple targets [91]—and can be even combined in a parallel array of isothermal nucleic acid amplification reactors, where the isolated nucleic acid is distributed to individual reaction chambers containing different pathogen-specific primers [34]. As such, the LAMP assay reaches 10~100 times higher sensitivity than classical PCR assays and overcomes susceptibility to potential enzymatic inhibitors present in food matrices. LAMP, however, induces high complexity in multiplex assays, due to the need for multiple primers per assay [92]. Both LAMP and RPA have been tested in paper-based formats [93,94], as well as in microfluidic devices [28,95]. However, efficient integrated commercial devices are still very limited. While the amplification reaction with either LAMP or RPA is extremely efficient, the quantification of the amplicons (amplification products) is still analytically difficult.

Other portable amplification methods have emerged in recent years. For instance, helicase-dependent amplification (HDA) is an isothermal DNA amplification method that uses an accessory protein DNA helicase to separate duplex DNA to single-stranded templates [96]. Subsequently, DNA polymerase enables primer extension. HDA was employed for development of a disposable device for the sensitive detection of toxigenic *C. difficile* [97], or for colorimetric detection of *Helicobacter pylori* DNA [98]. 

The rolling circle amplification (RCA) isothermal method uses a DNA or RNA polymerase nuclease enzyme to generate long single stranded DNA or RNA [99,100]. Thus, a nucleic acid sequence can be replicated hundreds of times in a short period, which significantly increase the sensitivity of the devices. During RCA, nucleotides are continuously added to a primer annealed to a circular template by polymerase, through which a single binding event can be amplified over a thousand-fold. RCA was successfully applied to quantitative multiplex detection of nucleic acid from plant pathogenic viruses and bacteria [101], *Salmonella* [102] or *L. monocytogenes* [103]. 

Strand displacement amplification (SDA) is an isothermal method that utilizes an endonuclease enzyme with a partially hybridized duplex DNA labeled with a fluorescein or a cationic-conjugated polyelectrolyte [104,105]. To undergo SDA, the target DNA sample is first cleaved with a restriction enzyme to create a double-stranded target fragment with defined 5′- and 3′-ends. The released double-stranded short fragments carrying fluorescein enables DNA detection through the monitoring of the cationic-conjugated polyelectrolyte or fluorescein fluorescence spectra. Coupling of SDA with a lateral flow aptasensor allowed the sensitive detection of *S. enteritidis* [106]. The authors used an aptamer specific to the outer membrane of *S. enteritidis* for magnetic bead pre-concentration, while a separate aptamer was used as a signal reporter for *S. enteritidis* for amplification by SDA prior to its detection by a lateral flow biosensor. Similarly, Wu et al. [107] incorporated aptamer-linked magnetic beads, for pre-concentration of *E. coli* O157:H7, to a lateral flow aptasensor which detected specific sequence amplified by isothermal SDA.

Nucleic acid sequence-based amplification (NASBA) is a sensitive, isothermal, transcription-based amplification system that can be used for the continuous amplification of RNA targets in a single mixture [108]. When applied as a diagnostic tool NASBA was shown to be quicker and more sensitive than PCR. NASBA was successfully applied to the detection and identification of mycobacteria [109], *L. monocytogenes* in dairy and egg products [110], *C. jejuni* in poultry products [111] or *S. enteritidis* in fresh meats, eggs, ready-to-eat salads and bakery products [112,113]. 

## 10. Paper-Based Detection of Nucleic Acid Sequences

Paper-based NATs provide an alternative to expensive and time-consuming conventional NATs, particularly when coupled with isothermal methods (see below). Recent advances in paper fabrication and modification technologies have made it possible to integrate all key steps of NATs (i.e., sample preparation, nucleic acid extraction and amplification and amplicon detection) into one single paper-based device [114]. 

Paper has been used as a modern analytical substrate since the mid-20th century, when it was used for chromatography and electrophoresis [115]. In the second half of the 20th century, nitrocellulose membrane emerged as a platform for the home pregnancy test. Nitrocellulose has remained a gold standard for lateral flow assays, and in general, for commercial diagnostics. In 2007, George Whitesides’s group from the Harvard University introduced paper as a 2D microfluidic diagnostic platform—i.e., microfluidic paper-based analytical device (μPAD) [116]. Ever since, paper-based diagnostics, particularly those for nucleic acid testing have been on the rise, due to several factors: (i) paper is low-cost and easy to acquire; (ii) paper is biocompatible; (iii) paper wicks fluids via capillary action and does not require external pumping sources; (iv) availability of papers with different physical properties (e.g., pore size, porosity, thickness, capillary flow rate etc.) for achieving different function of the assay; (v) paper can be easily modified (i.e., chemically-treated, cut, folded, stacked); (vi) paper can be safely disposed of by incineration, and vii) paper is scalable (i.e., amenable to printing and roll-to-roll manufacturing) [115,117]. Various paper-based formats are available, such as pH paper, nylon, chromatography paper for component separation, size-defined filters, dipsticks and lateral flow test, paper made of nitrocellulose, Dried Blood Spot cartridges, and Whatman FTA cards for the collection and storage of biological samples [31,114,118]. Further, a simple strategy can be adopted to stably immobilize oligonucleotides onto paper surfaces via ultraviolet irradiation.

Paper disks infused with the LAMP reagents and the primers of targeted genes were demonstrated for simultaneous, multiplex detection of foodborne pathogens [119]. The methodology uses 3,7,3′,4′-tetrahydroxyflavone, a bioactive plant flavonoid as an eco-friendly dye for interaction with DNA and fluorescence-based detection of pathogens (*E. coli* O157:H7, *Salmonella* spp., *S. aureus*, and *Cochlodinium polykrikoides*) in a single assay (Figure 5). A polycarbonate microdevice was fabricated, integrating paper-infused DNA extraction, LAMP, and on-chip detection modules, while the fluid flow was controlled with centrifugal force. Analysis of a real sample was performed with milk spiked with *Salmonella* spp. The purification of genomic DNA of foodborne pathogens was achieved after incubation of the heat-treated milk sample over paper doped with polydopamine, which reacted with milk ingredients and calcium ions and thus removed LAMP inhibitors. The microdevice showed high selectivity and LoD of approximately 170 cfu/mL in case of the spiked milk sample.

The first paper-based sensor using aptamer-initiated isothermal amplification via on-device RCA was also recently developed [120]. The sensing is based on two nitrocellulose disks connected through a physically connectable paper bridge. The first paper disk may contain a printed, fluorescently-labeled RNA or DNA aptamer-graphene oxide mixture, which act as a molecular recognition element. Upon the recognition of the pathogen analyte, the fluorescence is enhanced due to the aptamer detachment from graphene oxide surface. Upon connecting the bridge between the two paper discs, the analyte-aptamer conjugate migrates to the second disc containing RCA reagents for DNA amplification. The color is formed upon addition of the colorimetric assay reagents, which can be detected by the naked eye. The principle was effectively showcased by using an RNA aptamer on a paper device for the detection of ATP, a general bacterial marker and the use of a DNA aptamer for glutamate dehydrogenase, a marker for *C. difficile*. 

A highly sensitive and rapid point-of-need nucleic acid lateral flow assay has also been developed for the direct detection of RPA products [121]. For the lateral flow assay two biotinylated capture probes were immobilized on the test strip for each of the respectively test and control lines. On the test line, the immobilized probe is complemented with the 5′ end of the amplified DNA, while the control line binds to the control probe, which is conjugated to the gold nanoparticles for visual detection. The gold nanoparticle-conjugated control probe also binds to the 3′-end of the test line-attached amplified DNA and thus forms a sandwich, signaling the presence of the analyte DNA through appearance of the red color on the test line. The test line was observable at DNA concentration as low as 30 pM, with the LoD of 1 × 10^−11^ M using a smartphone camera and Image J software for detection, taking 15 min for the entire procedure. This combination of RPA, tailed primers, and a nucleic acid lateral flow system fulfills ASSURED requirements for point-of-need diagnostics.

Ahn et al. report a single-step RPA assay based on a paper chip manufactured by stacking functional papers [122]. The dry RPA reagents as well as the fluorescent probe were deposited on the reaction zone, comprised of a patterned poly(ether sulfone) membrane. Paper chip-based analysis showed optimal performance at 37 °C for 20 min with results comparable to those obtained with solution-based RPA. LoD achieved with this assay was 10^2^ cfu/mL with simultaneous detection of *E. coli*, *S. aureus* and *Salmonella typhimurium*. 

## 11. Microfluidics

Microfluidics, a technology of manipulating the small quantity of fluids in a network of microchannels, has found applications in various scientific and engineering disciplines including but not limited to, inkjet printing, chemistry, environment, and biomedicine. Currently, advanced microfluidics integrate into a single chip a number of operations, such as sample pre-treatment and preparation, cell separation, mixing and/or separation of fluids or cells together with micromechanical, optical, and electronic components for sensing and detection. The chip design can be optimized to accommodate on-chip storage of the isoNAT reagents, pre-loaded in stabilized form, as well as reservoirs for buffers liquids. The chambers can be designed for small volumes in order to economize reagents, facilitate rapid temperature control, and improve contrast of detection signals over background noise. Such design should result in cost reduction of the entire device, since the enzyme costs, which scale with reaction volume, can be a substantial (~50%) fraction of total chip cost [34]. In addition, such on-chip storage allows for more convenient use and reduces possible contamination and operator-associated errors [34]. LAMP and RPA reaction mixes are particularly useful since thy can be lyophilized for long shelf-life (>1 year) and stored in chips during manufacture or inserted from a library, prior to use [34].

To address advanced designing and manufacturing challenges of the microfluidic devices ready for the point-of-need testing, the research is mainly focused on: development and application of novel designs and novel materials, and application of advanced fabrication technologies for efficient prototyping and subsequent testing of the microfluidic chips. In order to integrate advanced functions into a single chip, various fabrication technologies were used for fabrication of microfluidic devices such as PolyDiMethylSiloxane (PDMS), Low Temperature Co-fired Ceramic (LTCC), 3D printing, xurographic technique, injection moulding, photolithography, X-ray lithography, laser ablation, micromachining, etc. [123,124]. The selection of the appropriate methodology depends on the application, chip complexity, applied detection principle, operating temperature, and many other factors. PDMS is widely used for the creation of organ-on-chip [125] or for point-of-need [126] devices due to good optical characteristics, flexibility, elasticity, and biocompatibility of the material. However, microfluidic chips fabricated using PDMS requires complex lithography process and manufacturing multilayered chips is a challenging task. To overcome those obstacles, techniques based on combining PDMS with SU-8 and quartz [127], lubricant-infused mould [128], and 3D printed mould [129] have been used for the chip fabrication. 

Complex multilayered microfluidic chips can be fabricated using LTCC technology [130,131]. LTCC combines the laser micromachining process for creation of complex microchannel geometries, screen printing process that allows deposition of conductive paste directly on LTCC tape, lamination and sintering processes. An important advantage of the LTCC is the possibility to create and test each layer separately before lamination. LTCC chips are characterized by good chemical and thermal stability, and very good mechanical properties. However, the drawback of LTCC technology is the nonblack of transparency, and therefore the additional binding of LTCC with glass or PDMS is required for the application where an optical detection principle is used. 

3D printing technology is another technology attracting significant attention in recent years in the production of microfluidics owing to its low-cost, simple fabrication process that can be performed in a single run, good system compatibility, and presence of a number of different materials with good biocompatible, chemical or mechanical properties. Different 3D printing processes have been used including Fuse Deposition Modeling (FDM), polyjet, electron beam melting, bioprint, and DLP-SLA printing techniques for the fabrication of the entire microfluidic device in a single run without the need for additional assembly processes [131,132,133,134]. Another advantage of 3D printing for the application in production of biosensors is the possibility to directly print different biomaterials, such as living cells or enzymes [133]. The potential of different 3D printing technology for microfluidic chip fabrication were compared in [135], while the utilization of 3D printing technology for applications in different microfluidic devices and sensors has been reported in a number of publications [131,132,133,135]. However, limitations of this process are low fabrication resolution and lack of transparency of materials which are often used in 3D processes. To improve optical properties of 3D printed microfluidic devices, some hybrid approaches were proposed such as multilateral 3D printing [136] or combined use of 3D printing materials and PDMS [129].

Furthermore, xurography can be used for rapid prototyping of low-cost, multi-layered microfluidic chips with good optical properties [137,138]. The xurographic technique combines cutting plotter process for cutting the channels in polymer foils and the lamination process for bonding a number of layers. Prior to lamination the screen-printing or inject-printing process can be used to print electrodes on the polymer. This rapid prototyping technique uses different polymers such as vinyl, polyvinyl chloride, polyethylene terephthalate, polyester, or polyimide. However, this technique suffers from uneven edges of the microchannels. Many other technologies or their combinations have been used for microfluidic chip fabrication, such as wet and dried etching, milling process, laser ablation, and many others. 

The selection of chip materials is largely depended on the fabrication process. The first generation of microfluidic devices which required complex fabrication processes was designed using glass, silicon or ceramic. Alternatively, elastomers enable low-cost, rapid prototyping and high chip integration, allowing complicated and parallel fluid manipulation. Plastics, as an alternative, for rapid and low-cost fabrication found an application in the number of microfluidic devices. In order to simplify fabrication processes and adapt them for industrial applications that require rapid fabrication of multilayered microfluidic devices with good performance, different materials have been developed for the applications in microfluidics. Additional polymers have been identified as complementary to PDMS in terms of transparency, biocompatibility and flexibility, but with higher rigidity and better resistance to solvents such as: thermoplastic elastomers [139], thermoset polyester [140], polyurethane methacrylate [141], and Norland Adhesive 81 [142]. Paper-based substrates provide an alternative to expensive polymers due to low-cost and ease of acquisition, biocompatibility, and availability of papers with different physical properties (porosity, thickness, capillary flow rate, etc.). Paper is particularly interesting for lateral flow assays, as it can wick fluids via capillary action and does not require external pumping sources. Based on these advances, paper-based microfluidic chips can integrate all key steps of NATs, such as sample preparation, nucleic acid extraction and amplification, and amplicon detection into same device [114]. 

Different bonding [143] and sealing techniques enable the construction of complex microfluidic devices that require sets of channels, chambers, valves and integrated micropumps, and with detection circuits yielding complete lab-on-a-chip device. In addition, when a multilayered chip design is used, techniques like thermal-pressure bonding [144], wax bonding [145], solvent bonding, ultrasonic welding [146], can be applied.

## 12. Detection Methods

Various signal read-out methods may be employed to detect amplified nucleic acids. Portable tests are based on signal transduction methods that are less equipment-intensive, such as optical (colorimetric and fluorimetric), electrochemical, microwave and magnetic. The transduction method is usually chosen for its sensory performance (in terms of limit of detection, dynamic range, and response time) in order to adapt it for on-site use (Figure 6). 

Most of the NAT diagnostic systems use optical or visual means to detect amplification products. Moreover, optical approaches are also the most common detection methods used in microfluidics. Optical detection methods include laser-induced fluorescence, absorbance, infrared/near-infrared and surface plasmon resonance, chemiluminescence, and colorimetric detection among others. External optical techniques and conventional optical instruments (including inverted fluorescence microscopy, CCD cameras, or light emission, and detector) are generally combined with a microfluidic system, mostly used for sample handling, separation or pre-treatment. However, the integration of optical detection equipment on a single chip to produce sensitive and compact microfluidic sensors for in-field detection is still challenging. 

The colorimetric method has often been applied to qualitative and semi-qualitative testing due to its direct observation by the naked eye. Colorimetric sensing is one of the most commonly used approaches for laboratory tests and industrial applications due to its advantages, such as low cost, easy integration with paper-based microfluidics, lack of external equipment, and the possibility to simultaneously detect a number of different analytes at the same time (multiplexing). After the sample is loaded and distributed over different reaction zones, analytes are detected using colorimetric assays by visually observing changes in color, intensity, brightness, or by measuring the amount of the reflected light from a surface caused by the presence of the analyte using detectors (typically cameras, smartphones or scanners) to quantify changes in color. In recent years, the colorimetric method has found a number of applications in: biochemical and medical analysis, forensic diagnostics, detection of glucose and protein concentration [116], toxins [147] and foodborne pathogen detection [148]. However, the main drawbacks of colorimetric detection include low sensor sensitivity and selectivity, as well as the possible release of toxic gases during reaction. In contrast, a colorimetric sensor may be successfully coupled with image recognition techniques, and mobile phone applications for quantitative analysis and interpretation of results [36,149]. 

The fluorescence-based method has been extensively used in microfluidic systems for labeling due to the wide variety of fluorescence detection labels available for targeting biomolecules. The popularity of this technique is likely due to the simplicity with which microfluidic devices can be coupled to fluorescence excitation and detection devices, as well as its high sensitivity and ability to detect pathogens in low sample volumes. Although microscope optics, CCDs, or photomultipliers (PMTs) commonly add substantial size and complexity to the detection systems, low-cost and simple devices for in-field detection can be designed using an LED for excitation and photodetector. Fluorescence detection has been widely used in the detection of bacteria and their toxins [150,151,152]. 

Chemiluminescence is the emission of light resulting from a chemical reaction, typically of specific substrate and an oxidant in the presence of cofactors. For instance, the targeted amplicon can be detected by the use of tag-specific antibody conjugated with horseradish peroxidase (HRP) enzyme (Figure 6). When the chromogenic substrate, such as luminol, is added, together with its activator H_2_O_2_, HRP catalyzes a reaction to release energy in the form of light. Emitted light can be easily detected. The advantage of this technique is that excitation light sources and emission filters are not required, which minimizes background interference. Several microfluidic devices that use the chemiluminescence detection method have been implemented in applications in medicine, blood analyses [153], but also detection of Staphylococcal Enterotoxin B100 [154]. Contrary to the use of fluorescence detection, chemiluminescence offers a simple detection method, which does not require complex instrumentation. However, the development of low-cost sensitive photodetectors is still necessary for the successful adoption of chemiluminescence microfluidic sensors in applications requiring disposable and easy-to-use devices. 

Electrochemical detection involves the interaction of chemical species with electrodes carrying an immobilized probe, such as DNA. Electrochemical detection on microchip platforms has become a popular technique for implementation in the field-portable devices [70,154,155,156,157,158,159,160]. Furthermore, electrochemical detection is one of the best alternatives to optical detection due to its inherent sensitivity, capability to be miniaturized without loss of performance, and high compatibility with microfabrication techniques. Electrochemical detection measures current, voltage, conductance, or impedance changes in the process of affinity bonding between receptor/ligand or antigen/antibody systems or enzyme catalyzed chemical reactions. An electrochemical sensor typically contains a simple electrode configuration, including a working electrode, a counter electrode, and a reference electrode. The functionalization of the electrode with enzymes can make use of their ability to selectively catalyze chemical reactions. Electrochemical detection offers a less expensive read-out than those implementing optical detection and can be easily miniaturized and integrated into microfluidic systems. Microfluidic, electrochemical detection has been successfully implemented in the detection of foodborne pathogens [5,70,155,156,161,162], and DNA encoding for bacterial toxins [163,164]. 

Microwave sensors are also suitable for food analysis since they operate in the frequency range of 0.3–300 GHz, which are non-destructive and safe, yet possessing an excellent detection potential even in the case of a small sample volume. In addition to good sensitivity and easy integration with different fabrication technologies and microfluidic, microwave sensors can be easily functionalized with enzymes or antibodies, making a favorable platform for in-field rapid detection. As such, microwave sensors have found numerous applications in permittivity sensing [130,154,165] and food quality control [166]. The operating principles of microwave sensors predominantly rely on the resonance concept, transmission or phase measurement, free space spectroscopy, or microwave imaging. Different configurations of the microwave resonators, transmission lines, microcantilevers or capacitors have been used as sensor elements for detection of cells, proteins or DNAs for various biological and environmental sensing applications [167,168]. 

Magnetic sensing is an encouraging alternative for the detection and measurement of different biological and chemical phenomena in life sciences, since it is a readily available technology, always combined with magnetic micro- and nanoparticle markers (Figure 7). This method involves the labeling of the biological entity with the magnetic particles and the detection of their stray field using highly sensitive magnetic sensors [157,169,170,171,172,173]. Current efforts are focused on the integration of such sensors within microfluidic platforms to develop simple, sensitive, and portable devices for rapid diagnosis of pathogens. Moreover, some of the magnetic sensors used for detection of the markers’ stray field are compatible with standard silicon integrated circuit technology, and thus suitable for integration into hand-held, portable, on-chip biosensing systems [171,174]. Sensing techniques based on magnetic particles have several advantages in terms of analytical figures of merit, such as high signal-to-noise ratio, high sensitivity, and fast analysis time [175,176]. By immobilizing additional biomolecules onto the magnetic particle surface, a number of additional functionalities emerge, such as transport of these biomolecules to a specific location for on-chip magnetic immunoseparation as well as measuring of biomolecular binding events. Two examples of magnetic biosensors are presented in Figure 7. Furthermore, the magnetic particle used for the labeling of the biological entity can be manipulated inside microfluidic channels by high gradient magnetic fields. Additional advantages of magnetic biosensing is the natural lack of any detectable magnetic content in biological samples which enables the development of sensing systems with low background noise, and therefore low LoD. There are two main magnetic sensing principles: the giant magnetoresistance (GMR) effect-based and magnetic particle quantification (MPQ)-based. 

GMR biosensors have emerged as excellent biodetection techniques at room temperature and as quantification methods of biological entities due to their high sensitivity, not very complex instrumentation, compact size, and integration flexibility [144,177,178]. The Lab-on-a-Chip magnetoresistive device commercialized by the Portuguese startup Magnomics (Lisbon, Portugal, Figure 7a) combines on-chip DNA extraction, amplification, and magnetic particle-based detection for bacterial detection, identification and antibiotic resistance profiling as a fast and portable solution [179,180]. The other principle, MPQ technique employs a non-linear magnetization of magnetic particles subjected to a magnetic field at AC frequencies f1 and f2 by recording the magnetic particle response at a combinatorial frequency f = n∙f1 + m∙f2, where n and m are integers (one of them can be zero). A MPQ-based biosensing platform was developed for rapid, high-precision, quantitative analyses for in vitro diagnostics by the Nikitin group (Figure 7b). This platform combines the merits of sandwich immunochromatography with highly sensitive quantification of magnetic particles from the entire volume of lateral flow membranes [179,180]. 

## 13. Read-Out Signal Amplification

In previous years, the most frequently applied labels to increase recognition signal intensity include enzymes such as alkaline phosphatase and horseradish peroxidase (Figure 6). However, in recent years, DNA biosensors have been coupled with metallic or semiconductor nanoparticles with unique optical or electrical properties to amplify intensity of the recognition signal [181]. Such labelling for detection of amplified DNA helps to increase assay stability and sensitivity. In the nanoparticle amplification mode, target DNA can be conjugated with nanoparticles, or unmodified target DNA is co-hybridized with oligonucleotides linked to nanoparticles, and the capturing oligonucleotide probe immobilized to the surface, in a sandwich format [182,183]. In addition, cross-linked gold nanoparticles immobilized on an electrode may provide a conductive matrix for DNA-associated methylene blue dye, and in this way amplify electrochemical detection of DNA [157,184]. Some examples of nanoparticle applications in enhancing DNA biosensor sensitivity for foodborne pathogen detection are given in Table 1. 

Laser irradiation may further enhance optical read-out signals by taking advantage of gold nanoparticle surface resonance [185]. For instance, 8-fold improved sensitivity of the gold nanoparticle-based lateral flow assay was achieved without increasing the device cost for the detection of *C. difficile* [186]. Finally, recent advances in nanotechnology, particularly concerning synthesis of nanocomposites with controlled physicochemical properties, offer novel particles with enhanced optical and electrochemical properties that can be applied in point-of-need biosensors [181,187]. For example, employing submicrobeads prepared by embedding numerous CdSe/ZnS quantum dots (QD), instead of a single QD, dramatically amplified the detection of aflatoxin B1 [188]. Indeed, the thousands of CdSe/ZnS quantum dots exhibited much brighter luminescence than single corresponding quantum dots. Similarly, recently it was shown that encapsulation of a great number of CdS QDs in ZIF-8 nanoparticles enabled a significant amplification of detected signal [189]. These CdS@ZIF-8 nanoparticles coated with specific anti-*E. coli* O157:H7 antibody detected *E. coli* with a LoD of 3 cfu/mL.

## 14. Data Management

As bionanotechnology is rapidly developing and the number of biosensor applications is rising, the use of big-data approaches for analytics of collected data is becoming increasingly needed. In the food safety sector, the use of point-of-need devices enables fast data collection, which poses a challenge for traditional approaches in data processing. Analyses at a large scale of food contaminations and outbreaks will help to achieve global conclusions, providing, in turn a powerful impact on future research directions and choices. In addition, early detection of food-borne pathogens could provide an important source of raw data needed to formulate predictive models of food contamination. Ultimately, predicting when and where certain food category is at risk of becoming contaminated is widely preferred instead of reacting *post festum* to outbreaks and contamination. With an ever-increasing scale of food production, the prediction of contamination possibilities may be necessary to decrease the risk of outbreaks as well as to reduce economic losses.

Accurate predictive models should incorporate diverse sources of data, including geographical, environmental, genetic origin of food, food production chain, and internet-based data sources. Big data mining is an approach facing Volume, Variety, Velocity and Value (4V) criteria which traditional data analysis is incapable of processing. Presently, similar approaches to the intelligent analysis of large-scale information are implemented in many biomedical, industrial and environmental monitoring categories. Taking into account that pathogen monitoring is an essential part of any pharmaceutical, medical, or biotechnological manufacturing processes and devices, microbial detection has the highest importance for the biotechnological sector. Data analysis at the large scale enables evidence-based knowledge discovery that can aid in various applications, ranging from the use of statistics to support the validity of the prevention strategies and treatment measures, to defining policy guidelines for food production and control. It is expected that predictive models will be soon based on not only dynamic data sources, but also on data acquired in real-time due to rapid development of IoT biosensor technologies for in situ applications. Implementation of data-driven decision support will enable taking protective actions sooner in at-risk areas and, thus will allow prevention of outbreaks, or increases in time to implement biosecurity measures. However, even though big data analytics shows great promise in scientific literature, its applications in real food safety practice are still rare. To enable the use of the strong potential of data mining and data modeling for pathogen detection and monitoring of food chain production, transport and storage, there is a need to develop multiplex pathogen detection devices, to combine them with a strong analytical technology, and to miniaturize them into a robust and friendly to use IoT devices.

## 15. Conclusions and Future Trends

To reach the end-users compliance and regulatory guidelines on food and water quality in an optimal manner two main challenges are addressed currently: (i) development of new bioassays for biomarker detection, and (ii) improvement of robustness of existing bioassays to adapt them for applications in-field, and/or with complex samples. The focus is largely placed on detection of the most prevalent bacterial pathogens, including *Campylobacter*, *Escherichia coli* and *Salmonella* serovars as well as on detection of bacterial toxins, all of which present a threat to public health and increase the risk of significant economic losses. Beyond the economic loss, food recalls cause significant damage to credibility and reputation of food brands. Since there can be a prolonged period in the food production chain, from food farming, production, processing, packaging, distribution, and consumption of raw and processed food products, any level of contamination may represent a great threat, thus, causing serious spoilage or even a disease outbreak due to rapid bacterial multiplication. Despite the immense importance of food analysis for microbiological hazards, on-site pathogen detection remains challenging due to several major difficulties: challenges in achieving reliable, repeatable and sensitive detection with acceptable accuracy levels.

Although novel point-of-need nucleic acid testing methods are being slowly adopted in the commercial space, portable and sensitive nucleic acid detection is still a growing field of research providing many exciting possibilities for food industry and regulatory policies. The challenges that are yet to be solved include adaptation of protocols and sensor performance in specific food systems, robust comparison against traditional technologies, and encouraging professional control analysts to replace traditional methods with the novel ones that offer significant benefits. 

## Figures and Tables

**Figure 1 sensors-19-01100-f001:**
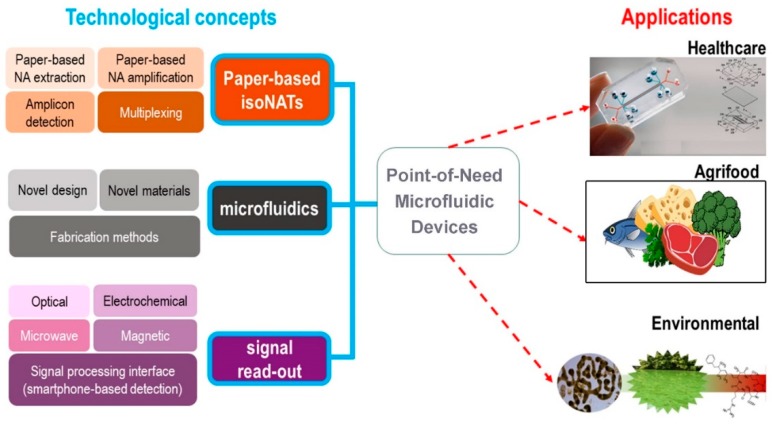
Scheme illustrating main technological concepts being developed in recent years as well as applications of portable low-cost biosensors coupled with isothermal nucleic acid testing (isoNAT) and microfluidics. NA, nucleic acid.

**Figure 2 sensors-19-01100-f002:**
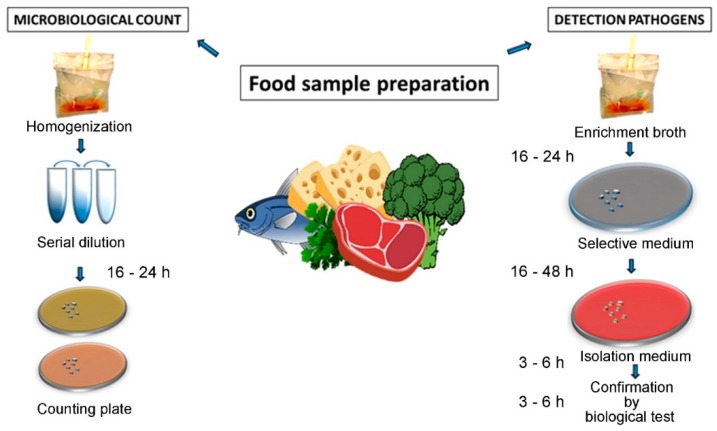
Classical detection methods based on bacterial cell counting and/or bacterial isolation on selective medium followed by biological test.

**Figure 3 sensors-19-01100-f003:**
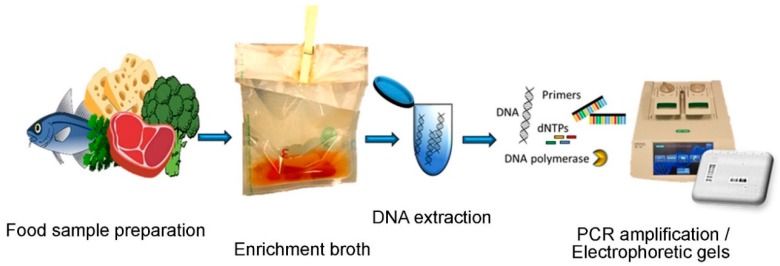
Molecular detection methods based on DNA extraction followed by biological test. dNTPs stands for deoxynucleotides.

**Figure 4 sensors-19-01100-f004:**
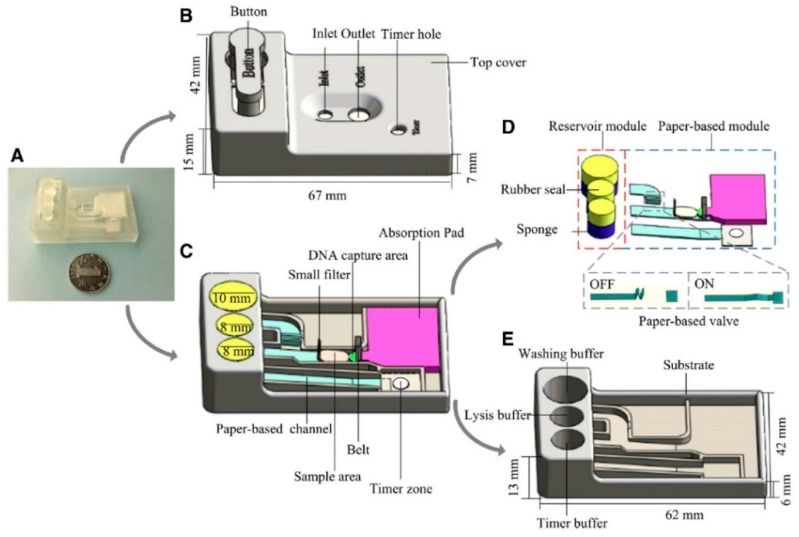
Paper-based DNA extraction device that couples sponge-based on-chip reagent storage, a valve and channels to autonomously direct the reagent and sample to the DNA capturing chip. (**a**) The photo image of paper-based DNA extraction device; (**b**) The top cover of the paper-based device; (**c**) The integrated platform of reservoir module and paper-based module supported by substrate; (**d**) The structure of reservoir module and paper-based module (including the paper-based valve); (**e**) The substrate. With permission from [63].

**Figure 5 sensors-19-01100-f005:**
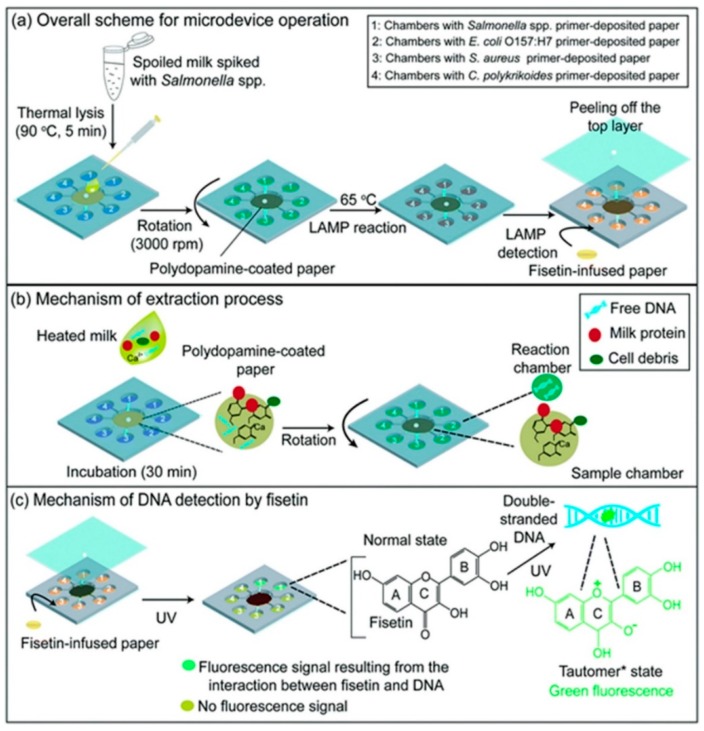
Schematic illustration demonstrating: (**a**) operation of the microdevice, (**b**) extraction process using heat-based lysis and polydopamine-coated paper, and (**c**) detection mechanism using fisetin. With permission from [119].

**Figure 6 sensors-19-01100-f006:**
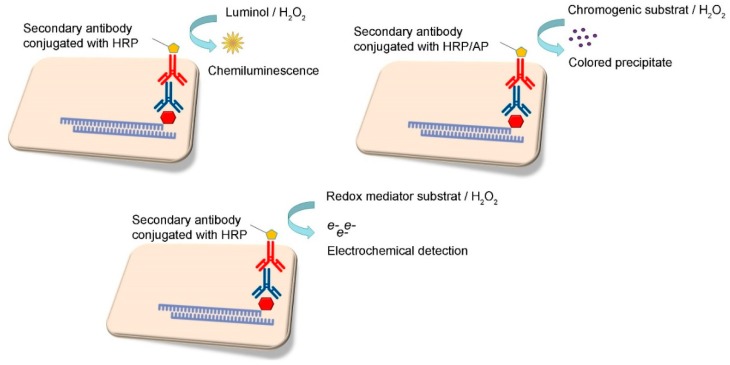
Tagged-amplicon detection by antibodies conjugated with the enzyme horseradish peroxidase (HRP) or alkaline phosphatase (AP) and their substrates via chemiluminescent, calorimetric, or electrochemical signal, respectively.

**Figure 7 sensors-19-01100-f007:**
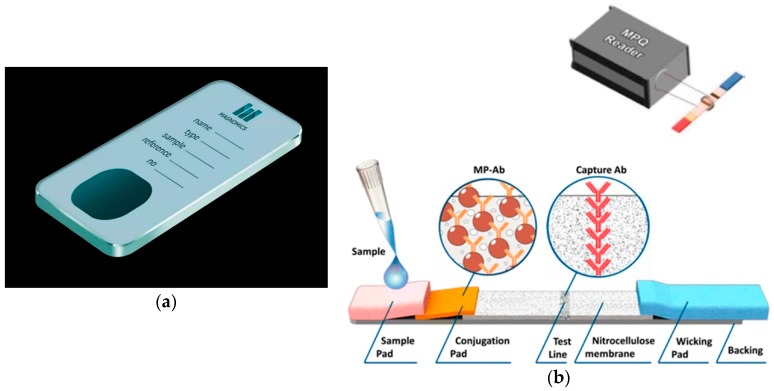
Examples of magnetic biosensors: (**a**) Device developed by Magnomics enables bacterial separation, DNA purification, PCR amplification and multiple pathogen GMR-based magnetic detection on the same chip. Image from www.magnomics.pt (**b**) Multiplex quantitative lateral flow assay for simultaneous point-on-care detection of different botulinum neurotoxin types realized via putting together a set of single-plex lateral flow strips, with magnetic nanolabels, a miniature cylinder cartridge and a portable multichannel reader based on the MPQ method. Adapted from [170].

**Table 1 sensors-19-01100-t001:** Examples of detection signal amplification by coupling biosensor with nanoparticles.

Detection Method	NP	Target	Receptor	Matrix	LOD	Reference
SEPR ^1^	Au	DNA	DNA probe			[190]
QCM ^2^	Au	*E. coli* O157:H7	DNA probe		2.0 × 10^3^ cfu/mL	[191]
Optical sensor	Fe_3_O_4_	*L. monocytogenes*	aptamer	milk	5.4 × 10^3^ cfu/mL	[192]
Voltammetry	Fe_3_O_4_	DNA	DNA probe		0.7 fmol	[193]
EIS ^3^/microfluidic	Ag	*E. coli*		Eggshell/ Tap water	500 cfu/mL	[194]
RF ^4^ sensor	Au	*E. coli*		Milk	10^5^ cfu/mL	[195]
DNA microarray	MNP ^5^	*E. coli*	DNA probe			[92]
		O157:H7,		Chicken meat	200 cfu/g	
		*S. enterica*,				
		*V. cholerae*				
		*C. jejuni*				
		*S. enterica*				
SPRI ^6^	Au	*S. aureus* *L. monocytogenes*	DNA probe		1 fM–1 attaM	[196]
SERS ^7^	Au	*E faecium*	DNA probe	Reference and Clinical samples	10 pM	[197]
		*S. aureus*				
		*S. maltophilia*				
		*V. vuiniculus*				

^1^ SEPR, surface-enhanced plasmon resonance; ^2^ QCM, piezoelectric biosensor; ^3^ EIS, electrochemical impedance; ^4^ RF, radio frequency; ^5^ MNP, magnetic nanoparticles; ^6^ SPRI, surface plasmon resonance imaging; ^7^ SERS, surface-enhanced Raman spectroscopy.

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
