# Peer review of "Point-of-Need DNA Testing for Detection of Foodborne Pathogenic Bacteria"

_sensors, 2019, doi:10.3390/s19051100_

Round 1
Reviewer 1 Report
I enjoyed reading this manuscript, it presents an exhaustive literature review about the state-of-the-art methods to for detection of foodborne pathogenic bacteria. All methods and techniques are very well discussed and they cover al most all exiting methods. The language is fluent and clear.
I found only the following minor issues:
- Page 3, figure 1. Specify in the legend the meaning of NA reported in some figure’s boxes, does it stand for Nucleic Acid?
- Page 3, line 92. I would suggest to add "selective" to ......culture media of a defined composition.....
- Page 4, line 129. TaqMan is the "nickname" of hydrolysis probes. I would suggest to mention that TaqMan are also called Hydrolysis probes or vice versa.
- Page 4, line 132. Replace "these" with "the" in the following sentence …. These molecular beacons are probes containing ……. "These" may confuse readers since indirectly references to the previous probe typology.
- Page 5, line 159. Give the meaning of STEC acronym
- Page 9, line 343. It might be useful to specify the meaning “cfu” at least once
Finally, I’d like to suggest to include, or at least mention, in section 4 or 9 or elsewhere, the DNA sequence method developed by Oxford Nanopore Technologies. This represent the ultimate portable system for fast DNA sequencing and is not based on PCR.
Author Response
Reviewer 1
I enjoyed reading this manuscript, it presents an exhaustive literature
review about the state-of-the-art methods to for detection of foodborne
pathogenic bacteria. All methods and techniques are very well discussed
and they cover al most all exiting methods. The language is fluent and
clear.
I found only the following minor issues:
- Page 3, figure 1. Specify in the legend the meaning of NA reported
in some figure’s boxes, does it stand for Nucleic Acid?
NA stands for Nucleic Acids. We have added the full name in the legend of Figure 1. P3, line 85
- Page 3, line 92. I would suggest to add "selective" to
......culture media of a defined composition.....
We agree with this remark as selective media explains the meaning of defined composition. Such media are various since depending on the microorganism and on the protocol to use. p. 3 line 92.
- Page 4, line 129. TaqMan is the "nickname" of hydrolysis probes. I
would suggest to mention that TaqMan are also called Hydrolysis probes or
vice versa.
We agree with the suggestion because “hydrolysis probes” can fit other probes that do not use Taq DNA polymerase. p. 4, line 129.
- Page 4, line 132. Replace "these" with "the" in the following
sentence …. These molecular beacons are probes containing ……. "These" may
confuse readers since indirectly references to the previous probe
typology.
We agree with this remark too. p. 4, line 132
- Page 5, line 159. Give the meaning of STEC acronym
We have modified “STEC” to “Shiga-toxin-Producing Escherichia coli (STEC)”, p. 5, line 160
- Page 9, line 343. It might be useful to specify the meaning “cfu”
at least once
We have added "colony forming units" before cfu p. 5, line 143, as it is the first time the cfu is mentioned in the paper.
Finally, I’d like to suggest to include, or at least mention, in section 4
or 9 or elsewhere, the DNA sequence method developed by Oxford Nanopore
Technologies. This represent the ultimate portable system for fast DNA
sequencing and is not based on PCR.
We have added a short explication on Oxford Nanopore Technologies and several examples of its applications, p.7, lines 266-276.
“In addition, the procedure can be complemented with on-site DNA sequencing using new technology for fast sequencing developed in 2014 by Oxford Nanopore Technology. This portable method is based on the utilization of a USB-powered sequencer (MinION) that comprises thousands of wells, each containing nanopores. When DNA enters the pore, each base produces a unique electronic signature that can be detected by the system, providing a readout of the DNA sequence.
The combination of isoNATs with portable DNA sequencing has been heavily investigated for human infectious diseases, such as malaria [43] or dengue virus [44] and is rapidly gaining attention for point-of-need detection of foodborne pathogens as well [45].”
Reviewer 2 Report
This review presents major developments achieved in recent years in point‐of‐need diagnostics in land‐based sector and sheds light on current challenges in achieving wider acceptance of portable devices in the food industry.
The topic is suitable for the publication on the Sensors Journal.
The abstract is complete and describes briefly the main and new developments for the detection of Foodborne Pathogenic Bacteria.
The manuscript is properly prepared.
The Tables, Figures and references are adequate for a review work.
In my opinion the manuscript should be accept after minor revision, adressing the following aspects:
Language improvement is recommended.
Please, the species names should be changed to cursive (Clostridium difficile, etc)
Line 819. Please, change the section number “4. Data management” by “14. Data management”.
Author Response
Reviewer 2
This review presents major developments achieved in recent years in point‐of‐need diagnostics in land‐based sector and sheds light on current challenges in achieving wider acceptance of portable devices in the food industry.
The topic is suitable for the publication on the Sensors Journal.
The abstract is complete and describes briefly the main and new developments for the detection of Foodborne Pathogenic Bacteria.
The manuscript is properly prepared.
The Tables, Figures and references are adequate for a review work.
In my opinion the manuscript should be accept after minor revision, adressing the following aspects:
Language improvement is recommended.
We thank you very much for your comments.
A native English speaker has checked the whole revised manuscript.
Please, the species names should be changed to cursive (Clostridium difficile, etc).
We have verified the whole manuscript for the bacterial names style.
Line 819. Please, change the section number “4. Data management” by “14.
Data management”.
Done.